# PointGPT: Auto-regressively Generative Pre-training from Point Clouds

Guangyan Chen[1]    Meiling Wang[1]    Yi Yang[1]    Kai Yu[1]    Li Yuan[2] *    Yufeng Yue[1] *

[1] Beijing Institute of Technology        [2] Peking University

## Abstract

Large language models (LLMs) based on the generative pre-training transformer (GPT) [46] have demonstrated remarkable effectiveness across a diverse range of downstream tasks. Inspired by the advancements of the GPT, we present PointGPT, a novel approach that extends the concept of GPT to point clouds, addressing the challenges associated with disorder properties, low information density, and task gaps. Specifically, a point cloud auto-regressive generation task is proposed to pre-train transformer models. Our method partitions the input point cloud into multiple point patches and arranges them in an ordered sequence based on their spatial proximity. Then, an extractor-generator based transformer decoder [27], with a dual masking strategy, learns latent representations conditioned on the preceding point patches, aiming to predict the next one in an auto-regressive manner. To explore scalability and enhance performance, a larger pre-training dataset is collected. Additionally, a subsequent post-pre-training stage is introduced, incorporating a labeled hybrid dataset. Our scalable approach allows for learning high-capacity models that generalize well, achieving state-of-the-art performance on various downstream tasks. In particular, our approach achieves classification accuracies of 94.9% on the ModelNet40 dataset and 93.4% on the ScanObjectNN dataset, outperforming all other transformer models. Furthermore, our method also attains new state-of-the-art accuracies on all four few-shot learning benchmarks. Codes are available at `https://github.com/CGuangyan-BIT/PointGPT`.

## 1   Introduction

Point clouds are becoming widely adopted data structures in various application areas, such as autonomous driving and robotics, emphasizing the importance of acquiring informative and comprehensive 3D representations. However, current 3D-centric approaches [9; 40; 41; 58; 32] typically necessitate fully-supervised training from scratch, which entails labor-intensive human annotations. In natural language processing (NLP) and image analysis domains, self-supervised learning (SSL) [11; 19; 47; 5; 46] has emerged as a promising approach for acquiring latent representations without relying on annotations. Among these methods, the generative pre-training transformer (GPT) [46] has been particularly effective at learning representative features [7], where the task is to predict data in an auto-regressive manner. Due to its remarkable performance, we naturally ask the question: can the GPT be adapted to point clouds and serve as an effective 3D representation learner?

To answer this question, we exploit the GPT scheme for point cloud understanding. However, it is challenging to employ GPT on point clouds due to the following reasons: (I) Disorder properties.

---
*Yufeng Yue and Li Yuan contributed equally to this study as co-corresponding authors. This work was supported by the National Natural Science Foundation of China under Grant No. NSFC 62003039, 62233002, 92370203, 61973034, the National Key RD Program of China (2022ZD0118), the CAST program under Grant No. YESS20200126.

37th Conference on Neural Information Processing Systems (NeurIPS 2023).

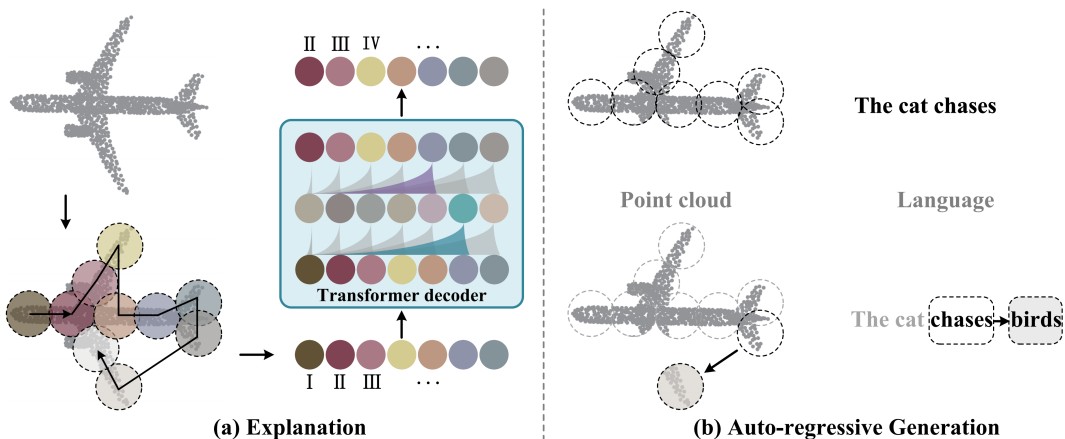

(a) Explanation  (b) Auto-regressive Generation

Figure 1: Illustration of our PointGPT. The transformer decoder is pre-trained to predict point patches in an auto-regressive manner. Such a design enables our method to predict patches without dedicated specifications and avoids positional information leakage, leading to improved generalization ability.

In contrast with the sequential arrangement of words in a sentence, a point cloud is a structure that lacks inherent order. To address this issue, point patches are arranged based on a geometric ordering, namely the Morton-order curve [36], which introduces sequential properties and preserves the local structures. (II) Information density differences. Languages are characterized by high information richness, therefore, the auto-regressive prediction task requires advanced language understanding. On the contrary, point clouds are natural signals with heavy redundancy, thereby the prediction task can be accomplished even without holistic comprehension. To address this disparity, a dual masking strategy is proposed, which additionally masks attending tokens for each token. This strategy effectively reduces redundancy and provides a challenging task that demands comprehensive understanding. (III) Gaps between generation and downstream tasks. Even though the model with a dual masking strategy exhibits sophisticated comprehension, the generation task primarily involves predicting individual points, which may result in the learned latent representations with a lower semantic level than downstream tasks. To mitigate this challenge, an extractor-generator architecture is introduced to the transformer decoder [27], such that the generation task is facilitated through the generator, thus enhancing the semantic level of the latent representations learned by the extractor.

Based on the above analysis, we propose a novel SSL framework for point clouds, called PointGPT. Specifically, our method partitions the input point cloud into multiple irregular point patches, which are subsequently organized using the Morton curve. Then, an extractor-generator based transformer decoder, with a dual masking strategy, processes the point patch sequences to learn latent representations conditioned on the unmasked preceding contents, and predicts the next point patches in an auto-regressive manner. Unlike recently developed masked point modeling approaches [66; 39; 68] that rely on positional information to specify reconstruction regions, resulting in the leakage of the overall object shape, our concise design, as illustrated in Figure 1, effectively circumvents the leakage of positional information and yields an enhanced generalization ability. Consequently, our PointGPT surpasses other single-modal SSL methods with comparable model sizes.

Inspired by the promising performance exhibited by our PointGPT, we endeavor to investigate its scaling property and push its performance limit further. However, a significant challenge arises due to the limited scale of the existing public point cloud datasets compared to NLP and images. This dataset size disparity introduces potential overfitting concerns. To alleviate this and fully unleash the power of PointGPT, a larger pre-training dataset is collected by mixing various point cloud datasets, such as ShapeNet [6] and S3DIS [3]. Moreover, a subsequent post-pre-training stage [57] is introduced, which involves performing supervised learning on the collected labeled dataset, enabling PointGPT to incorporate semantic information from multiple sources. Within this framework, our scaled models achieve state-of-the-art (SOTA) performance on various downstream tasks. In object classification tasks, our PointGPT achieves 94.9% accuracy on the ModelNet40 dataset and 93.4% accuracy on the ScanObjectNN dataset, outperforming all other transformer models. In few-shot learning tasks, our method also attains new SOTA performance on all four benchmarks.

Our main contributions can be summarized as follows: (I) A novel GPT scheme, termed PointGPT, is proposed for point cloud SSL. PointGPT leverages a point cloud auto-regressive generation task while mitigating positional information leakage, outperforming other single-modal SSL methods. (II) A dual masking strategy is proposed to create an effective generation task, and an extractor-generator transformer architecture is introduced to enhance the semantic level of the learned representations. These designs boost the performance of PointGPT on downstream tasks. (III) A post-pre-training stage is introduced, and larger datasets are collected to facilitate the training of high-capacity models. With PointGPT, our scaled models achieve SOTA performance on various downstream tasks.

## 2 Related Work

### 2.1 Self-supervised Learning for NLP and Image Processing

Self-supervised learning has attracted significant attention in recent years, especially in the fields of NLP and image processing, owing to its ability to learn useful representations without labeled data. The core idea of SSL is to design a pretext task to learn the distribution of the given data, obtaining beneficial features for the subsequent supervised modeling tasks [23; 13]. Contrastive learning [8; 16; 65; 38; 34; 44; 51] has been a popular discriminative self-supervised approach in both NLP and image processing, with the goal of grouping similar samples closer and diverse samples further apart. However, generative SSL methods [4; 19; 7; 11; 49; 46] have recently achieved more competitive performance. BERT [11] utilizes a bidirectional transformer to process the randomly masked text and reconstruct the original context. ELMo [49] adopts bidirectional LSTM [21] and generates subsequent words from left to right given representations of the previous contents. The GPT [46] also utilizes the auto-regressive prediction approach, but it employs a unidirectional transformer architecture, and the model is fine-tuned by updating all pre-trained parameters. In the computer vision field, BEiT [4] and MAE [19] randomly mask input patches, and pre-train models to recover the masked patches in the pixel space. Image-GPT [7] trains a sequence transformer to auto-regressively predict pixels without incorporating knowledge concerning the 2D input structure, exhibiting promising representation learning capabilities after pre-training.

### 2.2 Self-supervised Learning for Point Cloud

The success of SSL in NLP and image processing has motivated researchers to develop SSL frameworks for point cloud representation learning. Among these methods, the contrastive methods [61; 69; 37; 22; 63] have been extensively investigated. DepthContrast [69] constructs augmented depth maps and performs an instance discrimination task for the extracted global features. Similarly, MVIF [22] introduces cross-modal and cross-view invariance constraints to achieve self-supervised modal- and view-invariant feature learning. Another line of work [42; 12; 62] is proposed to integrate cross-modal information and leverage knowledge transferred from language or image models for 3D learning. ACT [12] employs cross-modal auto-encoders as teacher models to acquire knowledge from other modalities. Different from these methods, our work attempts to learn the intrinsic properties of point clouds without relying on cross-modal information and teacher models. Most relevant to our work are generative methods [24; 1; 50; 39; 66; 68; 33], especially recently proposed masked point modeling methods [39; 66; 68; 33]. Point-MAE extends the MAE by randomly masking point patches and reconstructing masked regions. Point-M2AE additionally utilizes a hierarchical transformer architecture and designs a corresponding masking strategy. However, masked point modeling methods still suffer from overall object shape leakage, which limits their ability to effectively generalize to downstream tasks. In this paper, we exploit the auto-regressive pre-training for point clouds and address unique challenges associated with the properties of point clouds. Our concise design avoids positional information leakage, thereby enhancing the generalization ability.

## 3 PointGPT

Given a point cloud $X = \{x_1, x_2, ..., x_M\} \subseteq \mathbb{R}^3$, the overall pipeline of PointGPT during pre-training is illustrated in Fig. 2. The point cloud sequencer module is utilized to construct an ordered sequence of point patches. This is achieved by dividing the point cloud into irregular patches and arranging them in Morton order. The resulting sequence is then fed into the extractor to learn latent representations, and the generator predicts the subsequent point patches in an auto-regressive manner. After the

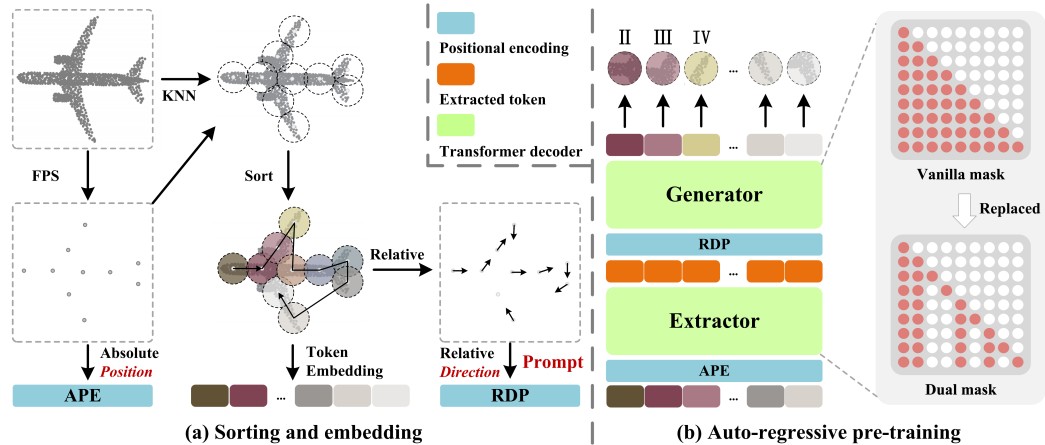

Figure 2: Overall architecture of our PointGPT. (a) The input point cloud is divided into multiple point patches, which are then sorted and arranged in an ordered sequence. (b) An extractor-generator based transformer decoder is employed along with a dual masking strategy for the auto-regressively prediction of the point patches. In this example, the additional mask of the dual masking strategy is applied to the same group of random tokens for better illustration purposes.

pre-training stage, the generator is discarded, and the extractor without the use of a dual masking strategy, leverages the learned latent representations for downstream tasks.

## 3.1 Point Cloud Sequencer

In the field of NLP, the GPT approach benefits from an easily accessible language vocabulary and the inherent ordered properties of words. In contrast, the point cloud domain lacks a predefined vocabulary, and a point cloud is a sparse structure that exhibits a characteristic of the disorder. To overcome these challenges and obtain an ordered point cloud sequence with each component unit capturing rich geometric information, a three-stage process consisting of point patch partitioning, sorting, and embedding is employed.

**Point patch partitioning**: Taking the inherent sparsity and disorder properties of point clouds into account, the input point clouds are processed by farthest point sampling (FPS) and the K-nearest neighbors (KNN) algorithms to obtain center points and point patches. Given a point cloud $\boldsymbol{X}$ with $M$ points, we initially sample $n$ center points $\boldsymbol{C}$ using FPS. Then, the KNN algorithm is utilized to construct $n$ point patches $\boldsymbol{P}$ by selecting the $k$ nearest points from $\boldsymbol{X}$ for each center point. In summary, the partitioning procedure is formulated as:

$$\boldsymbol{C} = \text{FPS}(\boldsymbol{X}), \quad \boldsymbol{C} \in \mathbb{R}^{n \times 3};$$
$$\boldsymbol{P} = \text{KNN}(\boldsymbol{C}, \boldsymbol{X}), \quad \boldsymbol{P} \in \mathbb{R}^{n \times k \times 3}. \quad (1)$$

**Sorting**: To address the inherent disorder properties of point clouds, the obtained point patches are organized into a coherent sequence based on their center points. Concretely, the coordinates of the center points are encoded into one-dimensional space using Morton code [36], followed by sorting to determine the order $\mathcal{O}$ of these center points. The point patches are then arranged in the same order. The sorted center points $\boldsymbol{C}^s$ and sorted point patches $\boldsymbol{P}^s$ are obtained as follows:

$$\mathcal{O} = \text{argmax}(\text{MortonCode}(\boldsymbol{C})), \quad \mathcal{O} \in \mathbb{R}^{n \times 1};$$
$$\boldsymbol{C}^s, \boldsymbol{P}^s = \boldsymbol{C}[\mathcal{O}], \boldsymbol{P}[\mathcal{O}], \quad \boldsymbol{C}^s \in \mathbb{R}^{n \times 3}, \boldsymbol{P}^s \in \mathbb{R}^{n \times k \times 3}. \quad (2)$$

**Embedding**: Following Point-MAE [39], a PointNet [40] network is employed to extract rich geometric information for each point patch. To facilitate training convergence, the normalized coordinates of each point are utilized with respect to its center point. Specifically, the sorted point patches $\boldsymbol{P}^s$ are embedded into $D$-dimensional tokens $\boldsymbol{T}$ as follows:

$$\boldsymbol{T} = \text{PointNet}(\boldsymbol{P}^s), \quad \boldsymbol{T} \in \mathbb{R}^{n \times D}. \quad (3)$$

## 3.2 Transformer Decoder with a Dual Masking Strategy

A straightforward extension of the GPT [46] to point clouds can be achieved by utilizing the vanilla transformer decoder to auto-regressively predict point patches, followed by fine-tuning all pre-trained parameters for downstream tasks. Nevertheless, this approach suffers from low-level semantics due to the limited information density of point clouds and the gaps between generation and downstream tasks. To address this issue, a dual masking strategy is proposed to facilitate comprehensive understanding of point clouds. Additionally, an extractor-generator transformer architecture is introduced, where the generator is more specialized for the generation task and is discarded after pre-training, enhancing the semantic level of the latent representations that are learned by the extractor.

**Dual masking strategy**: The vanilla masking strategy in the transformer decoder enables each token to receive information from all the preceding point tokens. To further encourage the learning of useful representations, the dual masking strategy is proposed, which additionally masks a proportion of the attending preceding tokens of each token during pre-training. The resulting dual mask $M^d$ is illustrated in Fig. 2(b), the self-attention process with a dual masking strategy can be represented as:

$$\text{SelfAttention}(\boldsymbol{T}) = \text{softmax}(\frac{\boldsymbol{Q}\boldsymbol{K}^T}{\sqrt{D}} - (1 - \boldsymbol{M}^d) \cdot \infty)\boldsymbol{V}. \quad (4)$$

where $\boldsymbol{Q}, \boldsymbol{K}, \boldsymbol{V}$ are $\boldsymbol{T}$ encoded with different weights for the $D$ channels. The masked locations in $M^d$ are set to 0, while the unmasked locations are set to 1.

**Extractor-generator**: Our extractor is composed entirely of transformer decoder blocks with a dual masking strategy, obtaining latent representations $\mathcal{T}$, where each point token only attends to the unmasked preceding tokens. Considering that point patches are represented in normalized coordinates, and global structures of point clouds are essential for point cloud understanding, sinusoidal positional encodings [55] (PE) are utilized to map the coordinates of the sorted center points $\boldsymbol{C}^s$ to the absolute positional encoding ($APE$). Positional encodings are added to every transformer block to provide location information and incorporate global structural information.

The generator architecture is similar to the extractor architecture but contains fewer transformer blocks. It takes the extracted tokens $\mathcal{T}$ as input and generates point tokens $\boldsymbol{T}^g$ for the following prediction head. However, the patch order may be affected by the center point sampling process, inducing ambiguity when predicting the subsequent patches. This hinders the model from effectively learning meaningful point cloud representations. To address this issue, the directions relative to the subsequent point patches are provided in the generator, serving as prompts without revealing the locations of the masked patches and the overall object shapes of point clouds. The relative direction prompts $RDP$ are formulated as:

$$RDP_i = \text{PE}((\boldsymbol{C}^s_{i+1} - \boldsymbol{C}^s_i)/\|\boldsymbol{C}^s_{i+1} - \boldsymbol{C}^s_i\|_2), i \in \{1, ..., n'\}, \quad RDP \in \mathbb{R}^{n' \times D}, \quad (5)$$

where $n' = n - 1$. In summary, the procedure in the extractor-generator architecture is formulated as:

$$\mathcal{T} = \text{Extractor}(\boldsymbol{T} + APE), \quad \mathcal{T} \in \mathbb{R}^{n \times D};$$
$$\boldsymbol{T}^g = \text{Generator}(\mathcal{T}_{1:n'} + RDP), \quad \boldsymbol{T}^g \in \mathbb{R}^{n' \times D}. \quad (6)$$

**Prediction head**. The prediction head is utilized to predict the subsequent point patches in the coordinate space. It consists of a two-layer MLP with two fully connected (FC) layers and rectified linear unit (ReLU) activation. The prediction head projects tokens $\boldsymbol{T}^g$ to vectors, where the number of output channels equals the total number of coordinates in a patch. Then, these vectors are reshaped to construct the predicted point patches $\boldsymbol{P}^{pd}$:

$$\boldsymbol{P}^{pd} = \text{Reshape}(\text{MLP}(\boldsymbol{T}^g), \quad \boldsymbol{P}^{pd} \in \mathbb{R}^{n' \times k \times 3}. \quad (7)$$

## 3.3 Generation Target

The generation target for each point patch is to predict the coordinates of the points within the subsequent point patches. Given the predicted point patches $\boldsymbol{P}^{pd}$, as well as the ground-truth point patches $\boldsymbol{P}^{gt}$, which correspond to the last $n'$ patches among the sorted point patches $\boldsymbol{P}^s$, the generation loss $\mathcal{L}^g$ is formulated using the $l_1$-form and $l_2$-form of the Chamfer distance (CD) [14], denoted as $\mathcal{L}^g_1$ and $\mathcal{L}^g_2$, respectively. Specifically, the generation loss is computed as $\mathcal{L}^g = \mathcal{L}^g_1 + \mathcal{L}^g_2$.

The $l_n$-form CD loss $\mathcal{L}_n^g$, with $n \in \{1, 2\}$, is defined as:

$$\mathcal{L}_n^g = \frac{1}{|\boldsymbol{P}^{pd}|} \sum_{a \in \boldsymbol{P}^{pd}} \min_{b \in \boldsymbol{P}^{gt}} \|a - b\|_n^n + \frac{1}{|\boldsymbol{P}^{gt}|} \sum_{b \in \boldsymbol{P}^{gt}} \min_{a \in \boldsymbol{P}^{pd}} \|a - b\|_n^n, \tag{8}$$

where $|\boldsymbol{P}|$ is the cardinality of the set $\boldsymbol{P}$ and $\|a - b\|_n$ represents the $L_n$ distance between $a$ and $b$.

We additionally find that incorporating the generation task into the fine-tuning process as an auxiliary objective can accelerate training convergence and improve the generalization ability of supervised models. This approach yields enhanced performance on downstream tasks, which is in line with the GPT [46]. Specifically, we optimize the following objective during the fine-tuning stage: $\mathcal{L}^f = \mathcal{L}^d + \lambda \times \mathcal{L}^g$, where $\mathcal{L}^d$ represents the loss for the downstream task, $\mathcal{L}^g$ represents the generation loss as previously defined, and the parameter $\lambda$ balances the contribution of each loss term.

### 3.4 Post-Pre-training

Current point cloud SSL methods directly fine-tune pre-trained models on the target dataset, which may result in potential overfitting due to the limited semantic supervision information [57]. To alleviate this issue and facilitate training of high-capacity models, we adopt the intermediate fine-tuning strategy [57; 4; 29] and introduce a post-pre-training stage for PointGPT. In this stage, a labeled hybrid dataset is leveraged (Sec. 4.1), which collects and aligns multiple point cloud datasets with labels. By conducting supervised training on this dataset, semantic information is effectively incorporated from diverse sources. Subsequently, fine-tuning is performed on the target dataset to transfer the learned general semantics to task-specific knowledge.

## 4 Experiments

This section begins by presenting the implementation and our pre-training setups. Subsequently, the effectiveness of our pre-trained models is evaluated across a range of downstream tasks. Finally, ablation studies are conducted to analyze the main properties of our PointGPT.

### 4.1 Implementation and Pre-training Setups

**Models**: Following previous studies [39; 66], PointGPT is trained employing the ViT-S configuration [67] for the extractor module, referred to as PointGPT-S. Additionally, we investigate the high-capacity models by scaling the extractor to the ViT-B and ViT-L configurations, denoted as PointGPT-B and PointGPT-L, respectively. More details can be found in the appendix.

**Data**: PointGPT-S is pre-trained on the ShapeNet [6] dataset without subsequent post-pre-training. This is in line with the previous SSL methods [39; 66; 68; 26] to allow for a direct comparison with these prior approaches. ShapeNet contains over 50,000 unique 3D models across 55 object categories. Additionally, two datasets are collected to support the training of high-capacity PointGPT models (PointGPT-B and PointGPT-L): (I) an unlabeled hybrid dataset (UHD) for self-supervised pre-training, which collects point clouds from various datasets [52; 35; 6; 53; 3; 60; 17], such as ShapeNet [6], S3DIS [3] for indoor scenes, and Semantic3D [17] for outdoor scenes, etc. In total, the UHD contains approximately 300K point clouds; (II) a labeled hybrid dataset (LHD) for supervised post-pre-training, which aligns the label semantics of different datasets [52; 35; 6; 53; 3; 60], with 87 categories and approximately 200K point clouds in total. Further details are provided in the appendix.

**Pre-training setups**: The input point clouds are obtained by sampling 1024 points from each raw point cloud. Afterward, each point cloud is partitioned into 64 point patches, with each patch consisting of 32 points. The PointGPT model is pre-trained for 300 epochs using an AdamW optimizer [31] with a batch size of 128, an initial learning rate of 0.001, and a weight decay of 0.05. Additionally, based on our empirical results, cosine learning rate decay [30] is employed.

### 4.2 Downstream Tasks

To demonstrate the performance of our method on different downstream tasks, we conduct experiments involving object classification on real-world and clean object datasets, few-shot learning, and part segmentation. The performance of PointGPT is evaluated using three different model capacities: PointGPT-S, which is pre-trained on the ShapeNet dataset without post-pre-training; and PointGPT-B

Table 1: Classification results on ScanObjectNN and ModelNet40 datasets. All results are expressed as percentages. Specifically, three variants are evaluated on the ScanObjectNN dataset. Additionally, The accuracy obtained on the ModelNet40 dataset is reported for both 1k and 8k points. The symbols ● and ● denote **larger pre-training dataset** and **post-pre-training stage**, respectively.

| Methods | Reference | ScanObjectNN | | | ModelNet40 | |
| --- | --- | --- | --- | --- | --- | --- |
| | | OBJ_BG | OBJ_ONLY | PB_T50_RS | 1k P | 8k P |
| *Supervised Learning Only* | | | | | | |
| PointNet [40] | CVPR 2017 | 73.3 | 79.2 | 68.0 | 89.2 | 90.8 |
| DGCNN [58] | TOG 2019 | 82.8 | 86.2 | 78.1 | 92.9 | - |
| PointCNN [25] | Neurips 2018 | 86.1 | 85.5 | 78.5 | 92.2 | - |
| GBNet [45] | TMM 2021 | - | - | 81.0 | 93.8 | - |
| MVTN [18] | ICCV 2021 | 92.6 | 92.3 | 82.8 | 93.8 | - |
| PointMLP [32] | ICLR 2022 | - | - | 85.4 | 94.5 | - |
| PointNeXt [43] | Neurips 2022 | - | - | 87.7 | 94.0 | - |
| P2P-RN101 [59] | Neurips 2022 | - | - | 87.4 | 93.1 | - |
| P2P-HorNet [59] | Neurips 2022 | - | - | 89.3 | 94.0 | - |
| *with Self-Supervised Representation Learning* | | | | | | |
| Point-BERT [66] | CVPR 2022 | 87.4 | 88.1 | 83.1 | 93.2 | 93.8 |
| MaskPoint [26] | ECCV 2022 | 89.3 | 88.1 | 84.3 | 93.8 | - |
| Point-MAE [39] | ECCV 2022 | 90.0 | 88.2 | 85.2 | 93.8 | 94.0 |
| Point-M2AE [68] | Neurips 2022 | 91.2 | 88.8 | 86.4 | 94.0 | - |
| **PointGPT-S** | - | **91.6** | **90.0** | **86.9** | **94.0** | **94.2** |
| **PointGPT-B** ● ● | - | **95.8** | **95.2** | **91.9** | **94.4** | **94.6** |
| **PointGPT-L** ● ● | - | **97.2** | **96.6** | **93.4** | **94.7** | **94.9** |
| Methods using cross-modal information and teacher models | | | | | | |
| ACT [12] | ICLR 2023 | 93.3 | 91.9 | 88.2 | 93.7 | 94.0 |
| PointMLP+ULIP [62] | CVPR 2023 | - | - | 89.4 | 94.5 | 94.7 |
| ReCon [42] | ICML 2023 | 95.4 | 93.6 | 91.3 | 94.5 | 94.7 |

and PointGPT-L, which undergo both pre-training and post-pre-training stages on the collected hybrid datasets. The impact of post-pre-training is further investigated and discussed in the appendix.

**Object classification on a real-world dataset**: The performance of the proposed method on a real-world dataset is an important indicator of its practical applicability. Therefore, the pre-trained models are transferred to the ScanObjectNN dataset [54], which contains approximately 15,000 objects extracted from real-world indoor scans. The experiments are conducted under three different settings, OBJ-BG, OBJ-ONLY, and PB-T50-RS. The results are presented in Table 1, our PointGPT-S, which has similar capacities and training data to previous methods like Point-MAE, outperforms other single-modal SSL methods. Furthermore, even when compared to Recon and ULIP, which utilize cross-modal information and teacher models, our scaled PointGPT-B model achieves superior performance, and PointGPT-L achieves accuracy improvements of at least 1.8%.

**Object classification on a clean objects dataset**: The pre-trained models are evaluated on the ModelNet40 dataset [60], which includes 12,311 clean 3D CAD models, covering 40 categories. To conduct fair comparisons, the standard voting method [28] is used during testing, and the input point clouds exclusively contain coordinate information, without additional normal information provided. The experimental results are presented in Table 1, our PointGPT-S surpasses other single-modal SSL methods. Even in comparison with Recon and ULIP, our PointGPT-L achieves superior performance.

**Few-shot learning**: To intuitively demonstrate the generalization ability of our method, few-shot learning experiments are conducted on the ModelNet40 dataset without the post-pre-training stage. Following the protocols of previous studies [39; 66; 68], the few-shot learning experiments consist of four distinct tests, employing $w$-way, $s$-shot setting. Specifically, $w \in \{5, 10\}$ represents the number of randomly selected classes, and $s \in \{10, 20\}$ denotes the number of randomly sampled objects for each selected class. Each test is conducted with 10 independent trials. The results, as shown in Table 2, indicate that our method outperforms other methods in all tests, particularly in the 10-shot tests. This demonstrates the ability of PointGPT in acquiring knowledge for new tasks, even under the constraints of limited training data.

Table 2: Few-shot classification results on the ModelNet40 dataset. In each experimental setting, 10 independent trials are conducted, and the mean accuracy (%) is reported with its standard deviation. Symbol • denotes **larger pre-training dataset**, and the **post-pre-training stage** • is excluded.

| Methods | Reference | 5-way | | 10-way | |
|---|---|---|---|---|---|
| | | 10-shot | 20-shot | 10-shot | 20-shot |
| DGCNN [58] | TOG 2019 | 31.6±2.8 | 40.8±4.6 | 19.9±2.1 | 16.9±1.5 |
| OcCo [56] | ICCV 2021 | 90.6±2.8 | 92.5±1.9 | 82.9±1.3 | 86.5±2.2 |
| *with Self-Supervised Representation Learning* | | | | | |
| Point-BERT [66] | CVPR 2022 | 94.6±3.1 | 96.3±2.7 | 91.0±5.4 | 92.7±5.1 |
| MaskPoint [26] | ECCV 2022 | 95.0±3.7 | 97.2±1.7 | 91.4±4.0 | 93.4±3.5 |
| Point-MAE [39] | ECCV 2022 | 96.3±2.5 | 97.8±1.8 | 92.6±4.1 | 95.0±3.0 |
| Point-M2AE [68] | Neurips 2022 | 96.8±1.8 | 98.3±1.4 | 92.3±4.5 | 95.0±3.0 |
| **PointGPT-S** | - | **96.8±2.0** | **98.6±1.1** | **92.6±4.6** | **95.2±3.4** |
| **PointGPT-B** • | - | **97.5±2.0** | **98.8±1.0** | **93.5±4.0** | **95.8±3.0** |
| **PointGPT-L** • | - | **98.0±1.9** | **99.0±1.0** | **94.1±3.3** | **96.1±2.8** |
| Methods using cross-modal information and teacher models | | | | | |
| ACT [12] | ICLR 2023 | 96.8±2.3 | 98.0±1.4 | 93.3±4.0 | 95.6±2.8 |
| ReCon [42] | ICML 2023 | 97.3±1.9 | 98.9±1.2 | 93.3±3.9 | 95.8±3.0 |

Table 3: Part segmentation results on the ShapeNetPart dataset. The mean intersection over union (mIoU) is reported across all classes (Cls.) and all instances (Inst.). Symbols • and • denote **larger pre-training dataset** and **post-pre-training stage**, respectively.

| Methods | Reference | Cls. mIoU(%) | Inst. mIoU(%) |
|---|---|---|---|
| PointNet [40] | CVPR 2017 | 80.4 | 83.7 |
| PointNet++ [41] | Neurips 2017 | 81.9 | 85.1 |
| DGCNN [58] | TOG 2019 | 82.3 | 85.2 |
| PointMLP [32] | ICLR 2022 | 84.6 | 86.1 |
| *with Self-Supervised Representation Learning* | | | |
| PointContrast [61] | ECCV 2020 | - | 85.1 |
| CrossPoint [2] | CVPR 2022 | - | 85.5 |
| Point-BERT [66] | CVPR 2022 | 84.1 | 85.6 |
| Point-MAE [39] | ECCV 2022 | - | 86.1 |
| **PointGPT-S** | - | **84.1** | **86.2** |
| **PointGPT-B** • • | - | **84.5** | **86.5** |
| **PointGPT-L** • • | - | **84.8** | **86.6** |
| Methods using cross-modal information and teacher models | | | |
| ACT [12] | ICLR 2023 | 84.7 | 86.1 |
| ReCon [42] | ICML 2023 | 84.8 | 86.4 |

**Part segmentation**: The representation learning capability of our method is evaluated on the ShapeNetPart [64] dataset, which consists of 16881 objects across 16 categories. The point clouds are sampled into 2048 points, and the segmentation head [39] concatenates the extracted features from layers $\frac{1 \times td}{3}$, $\frac{2 \times td}{3}$, and $\frac{3 \times td}{3}$ of the extractor transformer blocks, with $td$ representing the depth of the extractor. Subsequently, average pooling, max pooling, and upsampling are utilized to generate features for each point and an MLP is applied for label prediction. The experimental results, displayed in Table 3, demonstrate the superior performance of our PointGPT-L compared to all other methods.

## 4.3 Ablation Studies

Comprehensive ablation studies are conducted to investigate the fundamental designs of our PointGPT model. The impacts of these designs are evaluated by reporting the accuracy achieved by fine-tuning the model on the ModelNet40 dataset for object classification. To provide an intuitive representation of the results, ablation studies are conducted using the PointGPT-S model, which is pre-trained on the ShapeNet dataset and directly fine-tuned on the target dataset without post-pre-training.

Table 4: Ablation experiments with PointGPT-S pertaining on the ShapeNet dataset. The fine-tuned accuracy (%) achieved without post-pre-training is reported. Default settings are marked in `gray`.

(a) Generator depth.

| Blocks | Acc. |
|---|---|
| 0 | 93.85 % |
| 2 | 94.08 % |
| 4 | 94.21 % |
| 6 | 94.24 % |

(b) Generation targets.

| Targets | Acc. |
|---|---|
| Coordinates | 94.21 % |
| FPFH | 94.13 % |
| PointNet | 94.31 % |
| DGCNN | 94.35 % |

(c) Generation during fine-tuning.

| Coef. | Acc. |
|---|---|
| 0 | 94.01% |
| 1 | 94.15% |
| 3 | 94.21% |
| 5 | 94.05% |

(d) Generation loss.

| Loss | Acc. |
|---|---|
| CD $l1$ | 93.66% |
| CD $l2$ | 94.13% |
| CD $l1+l2$ | 94.21% |

(e) Relative direction prompts.

| Case | Acc. |
|---|---|
| None | 93.69% |
| Absolute position | 94.06% |
| Relative direction | 94.21% |

(f) Dual masking strategy.

| Ratio | Acc. | Ratio | Acc. |
|---|---|---|---|
| 0 | 93.68% | 5 | 94.01% |
| 1 | 93.70% | 7 | 94.21% |
| 3 | 93.85% | 9 | 93.66% |

Table 5: Classification results on ScanObjectNN and ModelNet40 datasets. The symbols ● and ● denote **larger pre-training dataset** and **post-pre-training stage**, respectively.

| Methods | ScanObjectNN | | | ModelNet40 | | Params |
|---|---|---|---|---|---|---|
| | OBJ_BG | OBJ_ONLY | PB_T50_RS | 1k P | 8k P | |
| *with different point sorting methods* | | | | | | |
| PointGPT-S w/ KD-Tree | 89.0 | 88.6 | 81.3 | 93.2 | 93.4 | 19.5M |
| PointGPT-S w/ Hilbert | 89.2 | 89.0 | 84.3 | 93.4 | 93.6 | 19.5M |
| **PointGPT-S** | **91.6** | **90.0** | **86.9** | **94.0** | **94.2** | **19.5M** |
| *with larger pre-training datasets and post-pre-training stage* | | | | | | |
| Point-MAE-B ● ● [39] | 94.2 | 93.9 | 90.2 | 94.2 | 94.4 | 120.1M |
| Point-M2AE-B ● ●[68] | 95.2 | 94.3 | 91.2 | 94.3 | 94.5 | 77.5M |
| **PointGPT-B ● ●** | **95.8** | **95.2** | **91.9** | **94.4** | **94.6** | **82.1M** |

**Generator**: Table 4(a) investigates the effect of varying the generator depth. The results demonstrate that the extractor-generator architecture facilitates the learning of strong semantic representations, particularly when combined with a deep generator, resulting in improved performance overall. However, due to the computational complexity associated with the deep generator, a depth of 4 is selected as the default setting for the generator.

**Generation targets**: The generation objective is essential for enabling the model to learn the intrinsic characteristics of the given data. Table 4(b) exhibits four distinct generation targets, which can be categorized into two groups: one-stage targets that can be directly obtained, including point coordinates and FPFH [48], and two-stage targets that are extracted by a trained deep network, including PointNet [40] and the DGCNN [58]. The experimental results indicate that the use of handcrafted FPFH features leads to underperformance, which may be attributed to the overfitting of low-level geometric features. The variants with two-stage targets outperform the variant with point coordinate targets. However, the pre-training and inference processes of the teacher model inevitably incur an additional computational cost.

**Generation task in the fine-tuning stage**: The generation task is included as an auxiliary objective during the fine-tuning stage. Table 4(c) presents the obtained results when varying the coefficient $\lambda$ of the generation loss in the fine-tuning loss. The results signify that this auxiliary objective serves as a regularization term and improves the generalization ability of supervised models. Furthermore, the results suggest that as the coefficient increases, the accuracy achieved in the classification task exhibits an increasing trend followed by a decreasing trend, reaching its highest value when $\lambda = 3$.

**Generation loss**: Table 4(d) presents the performance of variants using different generation loss functions, including the $l1$-form CD loss, the $l2$-form CD loss, and the combination of both the $l1$- and $l2$-forms CD loss. The results demonstrate that the combination of the $l1$- and $l2$-forms achieves

superior performance. We analyze that the $l2$-form is more effective in guiding the network toward convergence and the $l1$-form has better sparsity, thus the combination of both forms is more effective.

**Relative direction prompts**: The effect of utilizing relative direction prompts is analyzed in Table 4(e). The variant utilizing relative direction prompts outperforms the variants using absolute positional encoding and excluding positional encoding. We hypothesize that this improvement stems from the ability of relative direction prompts to prevent the model from overfitting to the patch order, thus enhancing the performance of PointGPT in downstream tasks.

**Dual masking strategy**: We conduct an analysis on the impact of the dual masking strategy and search for the proper mask ratio, as shown in Table 4(f). Decreasing the mask ratio to 0 corresponds to employing the vanilla masking strategy. The results indicate that both excessive and insufficient masking ratios lead to a decline in performance. The experimental results demonstrate that the dual masking strategy is effective in promoting beneficial representation learning and enhancing the generalization ability of the pre-trained model.

**Point sorting methods**: We analyze various point sorting methods, conducting ablation studies on the ScanObjectNN and ModelNet40 datasets using PointGPT-S models. Performance comparisons with the widely used KD-tree sorting [15] and Hilbert code sorting [20] methods are summarized in Table 5. The results indicate the substantial superiority of Morton code sorting. This observed effectiveness can be attributed to Morton code sorting's proficient preservation of the adjacency relationship between points, where adjacent points in one-dimensional space are often proximate to each other in three-dimensional space.

**Comparisons using larger pre-training dataset and post-pre-training**: Experiments are conducted by re-training comparison methods using the ViT-B configuration with collected larger datasets and post-pre-training stage. For a fair comparison, we focus on re-training well-performing single-modal self-supervised methods, Point-MAE, and Point-M2AE. The results, presented in Table 5, illustrate the superior performance of our PointGPT over comparison methods, utilizing comparable training parameters and the same training data.

## 5    Conclusion

In this paper, we present PointGPT, a novel approach that extends the GPT concept to point clouds, addressing the challenges associated with disorder properties, information density differences, and gaps between the generation and downstream tasks. Unlike recently proposed self-supervised masked point modeling approaches, our method avoids overall object shape leakage, attaining improved generalization ability. Additionally, we explore a high-capacity model training process and collect hybrid datasets for pre-training and post-pre-training. The effectiveness and strong generalization capabilities of our approach are verified on various tasks, indicating that our PointGPT outperforms other single-modal methods with similar model capacities. Furthermore, our scaled models achieve SOTA performance on various downstream tasks, without the need for cross-modal information and teacher models. Despite the promising performance exhibited by PointGPT, the data and model scales explored for PointGPT remain several orders of magnitude smaller than those in NLP [5; 10] and image processing [67; 29] domains. Our aspiration is that our research can stimulate further exploration in this direction and narrow the gaps between point clouds and these domains.

## Acknowledgements

This work was also supported in part by the National Key RD Program of China under Grant 2022ZD0118101, Natural Science Foundation of China NO. 62202014, and Shenzhen Basic Research Program JCYJ20220813151736001.

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
