# OpenReview forum: "PointGPT: Auto-regressively Generative Pre-training from Point Clouds"
_NeurIPS.cc/2023/Conference — NeurIPS 2023 poster_

### Official Review · Reviewer_LJmd · 2023-06-30

**Soundness:** 3 good
**Presentation:** 2 fair
**Contribution:** 2 fair
**Rating:** 5
**Confidence:** 4

**Summary:**

This paper investigates self-supervised point cloud learning by introducing the GPT concept (e.g., point order) to the masked modeling framework. Specifically, the clustered point patches are arranged into ordered sequences based on spatial proximity. Then, the masked patches can be predicted without leaking their position (e.g., patch centers) in an auto-regressive manner. Experiments are conducted on object tasks.

**Strengths:**

1. In my view, the most attractive point is the enlarged training set, which brings significant performance improvement over the competitors. (cf. Tab 1). This justifies the importance of the pretraining dataset for point clouds.
2. Significant performance on object tasks (e.g., shape classification), mainly due to the enlarged pertaining dataset.
3. The introduced relative direction prompt via point order is also reasonable, which avoids the position leak of patch reconstruction.

**Weaknesses:**

1. Although the relative direction prompt via point order is highlighted, it brings marginal performance improvement (cf. Tab 4 (e)). However, the enlarged pretrained dataset contributes the most performance improvement and is understated. Therefore, in my view, a paper revision should be conducted.
2. The proposed dual masking strategy sets some attention values to zero, which is similar to the popular attention dropout (cf. attention is all your need). Please clarify their differences.
3. Besides the object-level tasks, more practical downstream tasks (e.g., detection and segmentation with scene input) are expected to be evaluated, especially when the scene datasets are used in the pretraining stage in this paper.

**Questions:**

See the weakness

**Limitations:**

Non

---

> ### Author Rebuttal · Authors · 2023-08-09
>
> **@Q1 - The effectiveness of PointGPT's designs**.
>
> **(1) Relative direction prompt is able to enhance the generalization ability at a negligible cost**. Ablation experiments are conducted in a **high-capacity model training scenario**, where superior pre-training generalization is required. We re-train PointGPT-B using absolute positional encoding (`PointGPT-B-A`) on ScanObjectNN (OBJ_BG, OBJ_ONLY, PB_T50_RS) and ModelNet40 (1k points) datasets, and report the accuracy, the number of parameters (Params), and total pre-training time (Time). The results indicate that adopting the relative direction prompt leads to noticeable performance improvement at a negligible cost. We will incorporate the ablation experiments and results in the revised version to clarify the effectiveness of the relative direction prompt.
>
> |Methods|OBJ_BG|OBJ_ONLY|PB_T50_RS|ModelNet40|Params|Pre-training time|
> -|-|-|-|-|-|-|
> |PointGPT-B-A|95.2|94.5|91.3|94.2|82.1M|75 hours|
> |PointGPT-B|95.8|95.2|91.9|94.4|82.1M|75 hours|
>
> **(2)** Additionally, we conducted more experiments and verify that **the improvements introduced by PointGPT-B/L are closely associated with the designs of PointGPT**, which effectively address the information leakage and enhance the generalization ability. Specifically, additional experiments are conducted by re-training the well-performing methods Point-MAE and Point-M2AE using the ViT-B configuration with UHD and LHD datasets (`Point-MAE-B` and `Point-M2AE-B`). Experimental results exhibit the **significant performance improvement achieved by PointGPT under the same utilization of large datasets and model parameters**, despite Point-M2AE benefiting from the multi-scale features. For a fair comparison, we exclude methods using cross-modal information and teacher models.
>
> **(3) In the revised article, we will emphasize the importance of both PointGPT's designs and the training datasets**. This includes emphasizing the role of large datasets in improving accuracy in the Abstract and Introduction, as well as visually showcasing the performance gains achieved through enlarging the dataset size in the paper.
>
> |Methods|OBJ_BG|OBJ_ONLY|PB_T50_RS|ModelNet40|Params|Pre-training time|
> |-|-|-|-|-|-|-|
> |Point-MAE-B|94.2|93.9|90.2|94.2|120.1M|73 hours|
> |Point-M2AE-B|95.2|94.3|91.2|94.3|77.5M|156 hours|
> |PointGPT-B|95.8|95.2|91.9|94.4|82.1M|75 hours|
>
>
> **@Q2 - Differences between the masking strategy and attention dropout**.
>
> **(1) The effect on information leakage**. Although both methods involve the exclusion of random regions, the regions masked by the masking strategy remain consistent across different attention layers, thereby preventing information leakage from masked regions. In contrast, attention dropout varies across different layers, allowing the information dropped out in one layer to still be learned in other layers, resulting in information leakage.
>
> **(2) The impact on attention weight computation**. The masking strategy involves setting `the elements of the mask matrix` $M^{d}$ to 0. This results in the corresponding position being assigned a value of $-\infty$ during the softmax calculation, entirely eliminating the influence of masked regions on attention weight computation. In contrast, attention dropout simply sets the `obtained attention weights` to 0, allowing dropped regions to retain their impact on the calculation of attention weights for visible regions.
>
> **(3) Masking strategy outperforms attention dropout significantly**. Ablation studies are conducted to directly exhibit the effect of masking strategy and attention dropout. We re-train the PointGPT-S model with the attention dropout (`Dropout`), and the accuracy on the ScanObjectNN (OBJ_BG, OBJ_ONLY, PB_T50_RS) and ModelNet40 (1k points) datasets is reported below. The results demonstrate the superior performance achieved by the masking strategy in comparison to the attention dropout.
>
> |Methods|OBJ_BG|OBJ_ONLY|PB_T50_RS|ModelNet40|
> |-|-|-|-|-|
> |Dropout|89.1|87.9|84.3|93.1|
> |Ours|91.6|90.0|86.9|94.0|
>
> **@Q3 - Practical downstream tasks**.
>
> Thanks for your suggestions! We evaluate the performance of PointGPT in the **object detection task on the nuScenes dataset** and observe its adaptability to this task with minor adjustments for handling large-scale point clouds. Specifically, object detection experiments are conducted on the nuScenes dataset. PointGPT utilizes 12 attention blocks, each equipped with 8 heads, 128 input channels, and 256 hidden channels. Pre-training and fine-tuning of PointGPT are performed using the nuScenes dataset, excluding the post-pre-training stage. We compare PointGPT with several well-performing methods and report the major official metrics, mean Average Precision (mAP), and nuScenes detection score (NDS).
>
> |Methods|Reference|mAP|NDS|
> |-|-|-|-|
> |VISTA [1]|CVPR 2022|63.0|69.8|
> |Focals Conv [2]|CVPR 2022|63.8|70.0|
> |Transfusion [3]|CVPR 2022|65.5|70.2|
> |PointGPT||66.8|71.3|
>
> The specific modifications are as follows: (1) Voxelization: Replacing KNN-based grouping with voxelization, considering each voxel as a point patch. (2) Shift window attention: Confining the receptive field of the attention within a window and employing the window shift operation. (3) Point sampling: Randomly sampling up to K points within each voxel as the reconstruction targets.
>
> [1] Deng, Shengheng, et al. "Vista: Boosting 3d object detection via dual cross-view spatial attention." Proceedings of the IEEE/CVF Conference on Computer Vision and Pattern Recognition. 2022.
>
> [2] Chen, Yukang, et al. "Focal sparse convolutional networks for 3d object detection." Proceedings of the IEEE/CVF Conference on Computer Vision and Pattern Recognition. 2022.
>
> [3] Bai, Xuyang, et al. "Transfusion: Robust lidar-camera fusion for 3d object detection with transformers." Proceedings of the IEEE/CVF conference on computer vision and pattern recognition. 2022.

---

> > ### Comment · Reviewer_LJmd · 2023-08-17
> >
> > Thanks for the clarification. The effectiveness of the relative direction prompt and the proposed masking strategy is well received. Given that I am still concerned about the marginal improvement (i.e., about 0.5% Acc on average in Tab of Q1(2)) when fairly compared with the STOA (e.g., Point-M2AE), I will slightly raise my score to Borderline Accept.

---

> > > ### Author Response · Authors · 2023-08-18
> > > **Thanks for your feedback!**
> > >
> > > Thanks for considering our responses and raising the score! We will update our revised paper according to our discussions. Thanks again for your insightful and constructive suggestions that improve paper quality. We will be happy to address any further questions or concerns about the work.

---

### Official Review · Reviewer_4dwU · 2023-07-04

**Soundness:** 3 good
**Presentation:** 3 good
**Contribution:** 3 good
**Rating:** 6
**Confidence:** 5

**Summary:**

This paper proposes PointGPT, a new self-supervised learning strategy for 3D representation learning. PointGPT follows the success of autoregressive pre-training paradigm in NLP and adapts it into 3D point clouds. With larger pre-training dataset and a post-pre-training stage, PointGPT achieves SOTA performance on different benchmarks.

**Strengths:**

1. It's good to introduce GPT-style pre-training into 3D tasks. Although this idea is straightforward, this paper is the first attempt to my best knowledge.

2. The motivation of GPT pre-training in 3D is also reasonable and well clarified (avoid shape information leakage).

3. The writing, equations, and figures are easy to follow.

**Weaknesses:**

1. The experiment tables should incorporate the comparison of learnable parameters during training. It's very important to know the parameters of 3D pre-training to better judge the contribution.

2. How about other methods with larger pre-training dataset and post-pre-training? This is also a necessary ablation study to the paper.

3. It's better to cite this related paper in 'Methods using cross-modal information and teacher models':
Learning 3D Representations from 2D Pre-trained Models via Image-to-Point Masked Autoencoders, CVPR 2023

**Questions:**

I think it's an interesting paper. Expect the author rebuttal to solve my concerns for increasing the rating.

**Limitations:**

Yes

---

> ### Author Rebuttal · Authors · 2023-08-09
>
> **@Q1 - Learnable parameters**.
>
> Thanks for your suggestions! We will provide the number of learnable parameters (Params) in the revised article. For a fair comparison, we do not consider methods that utilize cross-modal information and teacher models, and mainly present the number of model parameters for well-performing single-modal self-supervised models here.
>
> |Models|Point-BERT|Point-MAE|Point-M2AE|PointGPT-S|PointGPT-B|PointGPT-L|
> |-|-|-|-|-|-|-|
> |Params|25.5M|29.0M|15.3M|19.5M|82.1M|242.2M|
>
> **@Q2 - Larger pre-training dataset and post-pre-training**.
>
> **PointGPT outperforms comparison methods with comparative training parameters and the same training data**. Specifically, experiments are conducted by re-training Point-MAE and Point-M2AE using the ViT-B configuration with collected larger datasets and post-pre-training stage. For a fair comparison, we focus on re-training well-performing single-modal self-supervised methods, Point-MAE, and Point-M2AE (`Point-MAE-B` and `Point-M2AE-B`). The results, encompassing ScanObjectNN (OBJ_BG, OBJ_ONLY, PB_T50_RS), ModelNet40 (1k points), number of parameters (Params), and total pre-training time, are presented below, which demonstrate the superior performance with high-capacity models. Additionally, we observe that Point-M2AE requires twice the pre-training time compared to our PointGPT, mainly due to its hierarchical architecture and dedicated multi-scale masking strategy.
>
> |Methods|OBJ_BG|OBJ_ONLY|PB_T50_RS|ModelNet40|Params|Pre-training time|
> |-|-|-|-|-|-|-|
> |Point-MAE-B|94.2|93.9|90.2|94.2|120.1M|73 hours|
> |Point-M2AE-B|95.2|94.3|91.2|94.3|77.5M|156 hours|
> |PointGPT-B|95.8|95.2|91.9|94.4|82.1M|75 hours|
>
> **@Q3 - Related work**.
>
> Thanks for your suggestions! We will cite this related paper 'Learning 3D Representations from 2D Pre-trained Models via Image-to-Point Masked Autoencoders'. Additionally, we will include more recent articles to enhance the coverage of our related work.

---

> > ### Comment · Reviewer_4dwU · 2023-08-17
> >
> > Thanks for the rebuttal. It addresses most of my concerns. It looks like PointGPT has good parameter and training efficiency. I recommend the authors include these results (parameter comparison and -B results) in the revised version. I will raise the rating.
> >
> > Also, I expect the authors to cite and discuss two other related works in CVPR 2023.
> >
> > [1] Parameter is not all you need: Starting from non-parametric networks for 3d point cloud analysis. CVPR 2023
> >
> > [2] Meta Architecture for Point Cloud Analysis. CVPR 2023

---

> > > ### Author Response · Authors · 2023-08-18
> > > **Thanks for your feedback!**
> > >
> > > Thanks for upgrading your score and providing valuable feedback! The submitted results will be integrated into the revised version, and these related papers will be cited and discussed in the revised article.
> > >
> > > **(1)** In the first paper, an innovative non-parametric network, Point-NN, is proposed. Building upon this, the Point-PN network is introduced, achieving promising performance across diverse tasks with only a few learnable parameters.
> > >
> > > **(2)** The second paper proposes a unified framework called PointMeta, exploring appropriate practices for each component to derive a fundamental building block named PointMetaBase. The proposed approach improves accuracy while reducing computational complexity.
> > >
> > > These approaches introduce effective frameworks that yield promising performance across a range of tasks. However, they still necessitate fully-supervised training from scratch. In contrast, PointGPT is able to acquire latent representations without relying on annotations, achieving enhanced generalization in downstream tasks. Notably, **our PointGPT consistently demonstrates superior performance** across the majority of benchmark evaluations. This verifies the effectiveness of our autoregressive generative pre-training approach for point clouds, affirming its capacity to enhance generalization and accuracy in downstream tasks.
> > >
> > > Thanks again for your insightful and constructive suggestions that improve paper quality! We will be happy to address any further questions or concerns about the work.

---

### Official Review · Reviewer_5U9K · 2023-07-08

**Soundness:** 4 excellent
**Presentation:** 3 good
**Contribution:** 4 excellent
**Rating:** 7
**Confidence:** 5

**Summary:**

This paper propose an auto-regressively generative pre-training paradigm for point cloud feature encoding. By incorporating GPT, the disorder and low information density properties of point clouds are addressed. Besides GPT, a dual masking strategy is proposed to improve the pre-training performance. The proposed models achieves SOTA o downstream tasks.

**Strengths:**

1.	A novel GPT-style point cloud pre-training framework is proposed.

2.	Arranging point patches via Morton-order curve is effective.

3.	The performance of the PointGPT-L is extraordinary, significantly surpassing previous methods.


**Weaknesses:**

See questions.

**Questions:**

I think this is an excellent work. But I still have some questions:

1.	Why PointGPT outperforms PointMAE and PointM2AE?  I don’t understand what is ``overall object shape leakage`` in L109. In L163, PointGPT also utilizes position information to extract global structural information.

2.	The dual masking strategy seems that further introduce MAE to PointGPT. What’s the performance when using the masking strategy only and removing the GPT loss?

3.	The training of PointGPT-S is aligned to previous pretraining methods and PointGPT-S only presents minor improvement compared with Point-M2AE. The PointGPT-B and PointGPT-L are trained on larger datasets but the competitive methods Point-M2AE and Point-MAE are only trained on ShapeNet. Please provide an ablation study such as Point-M2AE+UHD+LHD to show that the large performance gain doesn’t trivially come from the larger training dataset.


**Limitations:**

The Morton-order curve may not be the optimal order.

---

> ### Author Rebuttal · Authors · 2023-08-09
>
> **@Q1 - Overall object shape leakage**.
>
> **(1) The overall object shape leakage is attributed to the positional encoding leakage of masked regions**. Point cloud data is constituted by the spatial positions of individual points. However, previous methods rely on introducing positional encoding into mask tokens to specify prediction regions, leading to the positional encoding of masked regions being leaked. Consequently, the model can effortlessly infer the overall shape. **(2)** In contrast to prior methods, **PointGPT entirely masks the positional encoding of masked regions**. To achieve this, PointGPT eliminates the need for mask tokens and their associated positional encoding by utilizing the auto-regressive prediction pattern and the relative direction prompt. Consequently, while positional encoding remains employed, our method ensures the complete masking of information from masked regions.
>
> **@Q2 - Using the masking strategy only and removing the GPT loss**.
>
> Thanks for your valuable feedback! We directly train the PointGPT-S model on the target datasets and maintain the dual masking strategy (`PointGPT-S-D`), as the removal of GPT loss would render pre-training infeasible. The experiments are performed on the ScanObjectNN (OBJ_BG, OBJ_ONLY, PB_T50_RS) and ModelNet40 (1k points) datasets, and the results are shown below.
>
> |Methods|OBJ_BG|OBJ_ONLY|PB_T50_RS|ModelNet40|
> |-|-|-|-|-|
> |PointGPT-S-D|86.2|86.1|79.8|92.3|
> |Ours|91.6|90.0|86.9|94.0|
>
> We believe that your question aims to investigate the effectiveness of the dual masking strategy and the GPT loss. Therefore, we additionally conducted the following experiments focusing on these aspects.
>
> **(1) Dual masking strategy experiments**. Experiments are conducted by varying the masking ratio on the ModelNet40 dataset, the results reveal that utilizing the dual masking strategy with an appropriate masking ratio significantly boosts accuracy.
>
> |Ratio|0%|10%|30%|50%|70%|90%|
> |-|-|-|-|-|-|-|
> |Acc.|93.68|93.70|93.85|94.01|94.21|93.66|
>
> **(2) GPT loss experiments**. We conducted two supplementary experiments. **(i) Removal of GPT loss** (`PointGPT-S-R`). The PointGPT-S model is directly trained on the target datasets without the pre-training stage and masking strategy, serving as the baseline for the PointGPT-S framework. The results reveal the significant accuracy enhancement achieved by the PointGPT pre-training stage. **(ii) Replacement of GPT pre-training with MAE pre-training** (`PointGPT-S-M`). The auto-regressive pre-training of PointGPT is replaced with Point-MAE's masking and reconstruction pre-training, while utilizing the dual masking strategy. The findings suggest that under the application of the dual masking strategy, PointGPT's pre-training method achieves superior generalization ability.
>
> |Methods|OBJ_BG|OBJ_ONLY|PB_T50_RS|ModelNet40|
> |-|-|-|-|-|
> |PointGPT-S-R|86.7|87.1|79.8|92.3|
> |PointGPT-S-M|89.6|88.3|81.7|92.5|
> |Ours|91.6|90.0|86.9|94.0|
>
> **@Q3 - Comparison with Point-M2AE**.
>
> **(1) PointGPT-S achieves an enhanced accuracy-time trade-off**. It adopts a more concise design compared to Point-M2AE, which incorporates hierarchical architecture and a specialized multi-scale masking strategy involving backtracking. Consequently, **PointGPT-S achieves a remarkable 50% reduction in pre-training time**. This achievement holds substantial practical significance, conserving considerable wall-clock time, particularly when dealing with larger datasets and model parameters. Furthermore, PointGPT-S outperforms Point-M2AE-S across all tasks, despite Point-M2AE benefiting from multi-scale features.
>
> |Methods|Params|Flops|Pre-training time|
> |-|-|-|-|
> |Point-M2AE-S|15.3M|4.0G|32.5 hours|
> |PointGPT-S|19.5M|2.2G|15.8 hours|
>
> **(2)** Additionally, to demonstrate **the superior performance of PointGPT with high-capacity models**. Experiments are conducted by re-training Point-MAE and Point-M2AE using the ViT-B configuration with UHD and LHD datasets (`Point-MAE-B` and `Point-M2AE-B`). The results are depicted below, encompassing ScanObjectNN (OBJ_BG, OBJ_ONLY, PB_T50_RS) and ModelNet40 (1k points). These results demonstrate that **PointGPT outperforms comparison methods with comparative training parameters and identical training data**, and the improvements introduced by high-capacity PointGPT models are closely linked with the designs of PointGPT.
>
>
> |Methods|OBJ_BG|OBJ_ONLY|PB_T50_RS|ModelNet40|Params|Pre-training time|
> |-|-|-|-|-|-|-|
> |Point-MAE-B|94.2|93.9|90.2|94.2|120.1M|73 hours|
> |Point-M2AE-B|95.2|94.3|91.2|94.3|77.5M|156 hours|
> |PointGPT-B|95.8|95.2|91.9|94.4|82.1M|75 hours|
>
> **@Q4 - Morton code sorting**.
>
> We advocate for Morton code sorting as the appropriate solution. **(1) Morton code sorting effectively preserves the adjacency relationship** between points, with adjacent points in one-dimensional space often being proximate to each other in three-dimensional space. Moreover, **(2) experimental results demonstrate the improved accuracy achieved by Morton sorting**. Ablation studies are performed on ScanObjectNN (OBJ_BG, OBJ_ONLY, PB_T50_RS) and ModelNet40 (1k points) datasets, using the PointGPT-S models. The performance is compared with the popular `KD-tree` sorting [1] and `Hilbert` code sorting [2] methods, indicating that Morton code sorting significantly outperforms other sorting methods. **In the future, we will strive to explore better sorting methods.**
>
> |Methods|OBJ_BG|OBJ_ONLY|PB_T50_RS|ModelNet40|
> |-|-|-|-|-|
> |KD-tree|89.0|88.6|81.3|93.2|
> |Hilbert|89.2|89.0|84.3|93.4|
> |Ours|91.6|90.0|86.9|94.0|
>
> [1] Gadelha, et al. "Multiresolution tree networks for 3d point cloud processing." Proceedings of the European Conference on Computer Vision (ECCV). 2018.
>
> [2] Hilbert, et al. "Über die stetige Abbildung einer Linie auf ein Flächenstück." Dritter Band: Analysis· Grundlagen der Mathematik· Physik Verschiedenes: Nebst Einer Lebensgeschichte (1935): 1-2.

---

### Official Review · Reviewer_EUQH · 2023-07-13

**Soundness:** 2 fair
**Presentation:** 3 good
**Contribution:** 2 fair
**Rating:** 5
**Confidence:** 3

**Summary:**

This paper proposed a point clouds pretraining method named PointGPT, which extends the generative pretraining approach of NLP to point clouds. With point patch partitioning and sorting, point embeddings are feed into a transformer decoder for autoregressive prediction. Besides, a dual masking strategy is proposed to enhance the learned representation. PointGPT is evaluated on several point cloud classifiaction, few-shot classification and part segmentation datasets with promising results.

**Strengths:**

- Extend generative pretraining into point clonds. pointGPT let us to rethink the feasibility of generative pretraining on point cloud tasks.
- A complete framework with point sequencer and dual masking strategy. Re-arranging point clouds into a sequence of tokens like natural text and RGB images is challenging. PointGPT gives one possible solution using Morton code. The dual masking strategy appears to be a combination of masked modeling and autoregressive pretraining.
- Promising results on a set of datasets including classfication, few-shot classification and part seggmentation.

**Weaknesses:**

- Is it reasonable to use Morton code to sort unordered point clouds? As shown in Fig.2, PointGPT forces the current point $i$ to predict the coordinates of a point sorted by Morton code. However, the point $i$ should be able to predict any point adjacent to its 3D space, rather than a specified point. So what are the advantages of using Morton code sorting as the prediction target compared to conventional left-to-right and top-to-bottom prediction?
- The real gain of PointGPT (PointGPT-S) compared with the previous masked modeling methods (Point-M2AE) is not significant. Although PointGPT-B/L show an improvement on these tasks, they use more parameters and more training data.
- The overall contribution is limited. PointGPT is inspired by generative pretraining in NLP, but it doesn't show us how effective the generative pretraining approach is on point cloud tasks compared to previous methods.

**Questions:**

See weaknesses.

**Limitations:**

Yes.

---

> ### Author Rebuttal · Authors · 2023-08-09
>
> **@Q1 - Morton code sorting**.
>
> **(1) Morton code sorting effectively preserves the adjacency relationships between points**, allowing points that are close in three-dimensional space to maintain adjacency after sorting. However, the left-to-right and top-to-bottom (`L2R&T2B`) sorting method struggles to achieve this, due to the sparsity of point clouds. Furthermore, **(2)** to **prevent each point patch from predicting a fixed patch**, data augmentations, like rotation and translation, are applied to introduce variations in the order of the sorted patches. Consequently, each patch is tasked with predicting different patches under various transformations, necessitating accurate predictions within its local neighborhood. Notably, **(3)** experimental findings illustrate that **Morton sorting achieves improved accuracy**. To intuitively validate the effectiveness of Morton sorting, we perform additional experiments employing the PointGPT-S model on ScanObjectNN (OBJ_BG, OBJ_ONLY, PB_T50_RS) and ModelNet40 (1k points) datasets. The performance is compared with `L2R&T2B`, as well as the widely adopted `KD-tree` sorting [1] and `Hilbert` code sorting [2] methods. The results indicate that Morton code sorting outperforms alternative sorting methods.
>
> |Methods|OBJ_BG|OBJ_ONLY|PB_T50_RS|ModelNet40|
> |-|-|-|-|-|
> |L2R&T2B|88.8|88.4|80.8|93.2|
> |KD-tree|89.0|88.6|81.3|93.2|
> |Hilbert|89.2|89.0|84.3|93.4|
> |Ours|91.6|90.0|86.9|94.0|
>
> Details: We implement L2R&T2B by dividing the point cloud into multiple 0.06x0.06x0.06 grids [3] and sorting the points according to their grid coordinates, with a primary sorting based on the x-axis, followed by the y-axis, and ultimately the z-axis.
>
> [1] Gadelha, et al. "Multiresolution tree networks for 3d point cloud processing." Proceedings of the European Conference on Computer Vision (ECCV). 2018.
>
> [2] Hilbert, et al. "Über die stetige Abbildung einer Linie auf ein Flächenstück." Dritter Band: Analysis· Grundlagen der Mathematik· Physik Verschiedenes: Nebst Einer Lebensgeschichte (1935): 1-2.
>
> [3] Thomas, et al. "Kpconv: Flexible and deformable convolution for point clouds." Proceedings of the IEEE/CVF international conference on computer vision. 2019.
>
> **@Q2 - Comparison with Point-M2AE**.
>
> **(1) PointGPT-S attains an improved accuracy-time balance**. PointGPT adopts a more concise design compared to Point-M2AE, which incorporates hierarchical architecture and a specialized multi-scale masking strategy involving backtracking. Consequently, **PointGPT demonstrates a remarkable 50% reduction in pre-training time**, which holds significant practical value, saving considerable wall-clock time when scaling up datasets and model parameters. Furthermore, PointGPT-S surpasses Point-M2AE-S across all tasks, despite Point-M2AE benefiting from multi-scale features. Importantly, PointGPT is not confined to small-scale models, we explore the potential of high-capacity models.
>
> |Methods|Params|Flops|Pre-training time|
> |-|-|-|-|
> |Point-M2AE-S|15.3M|4.0G|32.5 hours|
> |PointGPT-S|19.5M|2.2G|15.8 hours|
>
> **(2)** The additional experiments demonstrate that **the improvements introduced by PointGPT-B/L are closely linked with the designs of PointGPT**, which effectively address the information leakage and enhance the generalization ability. Specifically, experiments are performed to re-train Point-M2AE with ViT-B configurations using UHD and LHD datasets (`Point-M2AE-B`). The results are presented below, encompassing ScanObjectNN (OBJ_BG, OBJ_ONLY, PB_T50_RS), ModelNet40 (1k points), number of parameters (Params), and total pre-training time, demonstrating **the superior performance of PointGPT over Point-M2AE models, even under comparable training parameters and identical training data**.
>
> |Methods|OBJ_BG|OBJ_ONLY|PB_T50_RS|ModelNet40|Params|Pre-training time|
> |-|-|-|-|-|-|-|
> |Point-M2AE-B|95.2|94.3|91.2|94.3|77.5M|156 hours|
> |PointGPT-B|95.8|95.2|91.9|94.4|82.1M|75 hours|
>
> **@Q3 - Contributions**.
>
> Recent pre-training methods for point clouds also belong to the generative pre-training approach, such as Point-MAE. However, a foundational distinction lies in their adoption of BERT-style masked modeling pre-training approaches, contrasting with our exploration of GPT-style auto-regressively generative pre-training methods.
>
> **Compared to preceding methods, PointGPT effectively addresses the information leakage**. Previous methods rely on introducing positional information to specify the regions for prediction, which leads to a significant and widespread issue of information leakage, limiting the efficacy of the pre-training process. To overcome this challenge, PointGPT utilizes an auto-regressive pattern and the relative direction prompt to specify prediction patches. This approach obviates the need for explicitly utilizing positional information, effectively addressing information leakage, and enhancing the generalization ability, as demonstrated in @Q2.
>
> **Our contributions also encompass (1) the first attempt of GPT on point clouds and (2) the exploration of high-capacity model training** within the point cloud domain. (1) PointGPT explores the point cloud sorting methods to manage the disordered nature of point clouds. Additionally, we propose a dual masking strategy and an extractor-generator architecture to overcome the challenges associated with information density differences and gaps between the generation and downstream tasks. (2) To fully unleash the power of PointGPT, we collect a larger pre-training dataset. Moreover, a subsequent post-pre-training phase is introduced alongside a labeled hybrid dataset, facilitating the integration of semantic information from various sources into the models.
>
> |(1) First attempt of GPT on point clouds|(2) High-capacity model training|
> |-|--|
> |point cloud sorting|post-pre-training stage|
> |dual-masking strategy|unlabeled hybrid dataset|
> |extractor-generator architecture|labeled hybrid dataset|
> |relative direction prompt||

---

### Author Rebuttal · Authors · 2023-08-09

We thank all reviewers for their thoughtful feedback! We are pleased to find that reviewers 5U9K and 4dwU appreciate PointGPT as interesting and excellent work. Moreover, reviewers 4dwU and LJmd consider our motivation to be reasonable and well-explained, effectively mitigating information leakage. Furthermore, reviewers 4dwU and EUQH appreciate the novelty of our method as the first attempt of auto-regressive pre-training on point cloud tasks, prompting the rethinking of this approach in such tasks. We are delighted that all reviewers acknowledge the significant improvements and promising performance achieved by our approach. We have carefully considered all questions, concerns, and comments provided by reviewers and addressed all of them appropriately. We provide detailed responses to each review separately and believe that our responses address all of the reviewers' concerns.

---

### Decision · Program_Chairs · 2023-09-21

**Decision:**

Accept (poster)

**Comment:**

This paper proposed a point clouds pretraining method named PointGPT, which extends the generative pretraining approach of NLP to point clouds. With point patch partitioning and sorting, point embeddings are fed into a transformer decoder for autoregressive prediction. In addition, a dual masking strategy is proposed to enhance the learned representation. PointGPT is evaluated on multiple datasets and show good performance.

The authors are recommended to include all added experimental results in the rebuttal to EUQH, 5U9K, LJmd, 4dwU.